# Spatial Variations of Trace Metals and Their Complexation Behavior with DOM in the Water of Dianchi Lake, China

**DOI:** 10.3390/ijerph16244919

**Published:** 2019-12-05

**Authors:** Yuanbi Yi, Min Xiao, Khan M. G. Mostofa, Sen Xu, Zhongliang Wang

**Affiliations:** 1Institute of Surface-Earth System Science, Tianjin University, Tianjin 300072, China; yuanbi.yi@tju.edu.cn (Y.Y.); mostofa@tju.edu.cn (K.M.G.M.); xusen@tju.edu.cn (S.X.); 2Tianjin Key Laboratory of Water Resources and Environment, Tianjin Normal University, Tianjin 300387, China; wangzhongliang@vip.skleg.cn

**Keywords:** Dianchi Lake, water quality, trace metals, DOM, minerals, optical properties

## Abstract

The dynamics of trace metals and the complexation behavior related to organic matter in the interface between water and sediment would influence water quality and evolution in the lake system. This study characterized the distribution of trace metals and the optical properties of dissolved organic matter (DOM) on the surface, and the underlying and pore water of Dianchi Lake (DC) to understand the origin of metals and complexation mechanisms to DOM. Some species of trace metals were detected and Al, Ti, Fe, Zn, Sr and Ba were found to be the main types of metals in the aquatic environment of DC. Ti, Fe, Sr and Ba predominated in water above the depositional layer. Al, Ti, Fe and Sr were the most abundant metallic types in pore water. Mn and Zn were the main type found at the southern lake site, reflecting the contribution of pollution from an inflowing river. The correlations between DOM and metals suggested that both originated from the major source as particulate organic matter (POM), associated with weathering of Ca-, Mg-carbonate detritus and Fe- or Mn-bearing minerals. High dynamics of DOM and hydrochemical conditions would change most metal contents and speciation in different water compartments. Proportions of trace metals in dissolved organic carbon (DOC) in natural waters were correlated with both DOM molecular weight and structure, different metals were regulated by different organic properties, and the same metal also had specific binding characteristic with DOM in various water compartments. This study highlighted the interrelation of DOM and metals, as well as the pivotal role that organic matter and nutrients played during input, migrations and transformations of metals, thereby reflecting water quality evolution in the lake systems.

## 1. Introduction

It is important to delineate the behavior and sources of metals in sediments/suspended particulate matter and pollution level in the aquatic system [1,2,3,4]. Trace metals and organic matter interacted in a series of biogeochemical processes, and nutritive elements were not excluded. Oxidation of organic matter mediated by trace metals might be produced under conditions found in aquatic environments and the reduction of high-valence metals could be coupled to the cycling of Fe, S and C [5]. Poorly crystalline Fe (hydr)oxide transformation during multiple, short redox cycles can provide insight into reactions controlling the speciation of redox active nutrients and trace metals in lake sediments [6]. N, S and P as well as organic matter were included in these biogeochemical transformations and removals in the lakes [7].

In fact, metal oxide cycling was shown to be more important in organic carbon mineralization pathways than previously recognized, in which nutrients were anticipated and N, Si, P, S, etc. played key roles in determining metallic quota among different phases [8]. Autotrophic denitrification of nitrate by Fe (II) and Mn (II) oxidations happened under nitrate reductase gene and Mn(II)-oxide gene, with the result that nitrate was almost turned into N_2_, 43% of Mn(II) and 100% of Fe(II) were oxidized simultaneously [9]. The bound phase and distributions of different metal species in various aquatic environments differed from each other. U, V and Mo exhibited conservative behavior, while Ni, Cu, As, Co and Cd showed additive behavior in desorption processes in the Mekong Delta [10]. Cd and Zn enhanced settling with bio-detritus of upwelling waters, Pb, Co and Ni were evidenced to associate through adsorption of Fe-Mn oxides nodules onto clay minerals, whereas Cu precipitated as Cu sulfides during reducing reactions in the coast along Southwest India [11]. Cu and Pb have high affinity for particulate fraction, Ni and Zn dominate in soluble fraction, which is comprised of free metal ions, complexes and metals entrained in colloidal suspensions [11,12]. Deposited metals originated from bacterial cells encapsulated by clay minerals, or adsorption by oxyhydroxides/detrital clays to bring to sediment [13,14]. Once the deposition condition changed, these metals could be released into pore water, and grain size influenced greatly the trapping of metals in the solid phase [15,16].

Iron oxides, organic matter and amorphous clay minerals control the removing and transforming of trace metals in water; organic matter not only fixed trace metals as Ni, Cd, Cu and Zn into sediments, but also removed them [1,4,17]. Generally, humic substances are the primary metal-binding ligands for trace metals, and most O-containing acid functional groups, such as carboxyls and phenols and relatively low abundance S- and N-containing functional groups, were proved to be metal-binding ligands, and a molecular structure proximal to metal-binding sites may be significant in metal coordination by DOM, creating stable complexes of varying complexing strengths [18,19]. Cu and Fe exhibited higher affinity compared to Zn, which were highly associated with the optical properties of DOM in the Sagami River Basin and other freshwaters in temperate climates [20]. Functional groups in DOM formed complexation with trace metal ions by donation of electrons from functional groups (F:) into empty or partially filled s-, p- or d- orbitals in metal ions (M^n+^) through a strong π-electron bonding system (F:M^n+^) [17].

Trace metals were usually used to reconstruct the depositional environment of organic-rich deposits [21], but most of these studies have focused primarily on hydrographically restricted basins. The response of trace metals to changing redox conditions, and the source and characteristic of trace metals in the nutrient cycle in lakes remains relatively unexplored. The objective of this study is to gain insight into the role of organic matter and related nutrients on the cycling of trace metals, thereby analyzing the geological genesis of trace metals and complexing mechanisms acting on dissolved organic matter at the surface, underlying and pore water of Dianchi Lake (DC). The study of geochemical characteristics of major and trace metals in different water compartments could reveal the sources, sedimentary environment and pollution status of the area. Quantitative information of additive and removal processes of these trace metals is essential for a better understanding of aquatic biogeochemistry in lakes and their full utilization as geochemical proxies.

## 2. Materials and Methods

### 2.1. Study Area

Dianchi Lake (24°40’–25°02’ N, 102°36´–102°47´ E) is a tectonic lake situated in the southwest of China, in the middle part of Yunnan-Guizhou Plateau (Figure 1), the sixth largest inland freshwater lake in China. Quaternary sedimentary strata distributed in the Dianchi Lake area, including extensive carbonate mineral. It covers a watershed area of 2920 km^2^ with a lake area of 309.5 km^2^, average depth of water is 4.4 m with impoundage of 15.7 × 10^8^ m^3^. It provides water for industrial and agricultural production, regulation and storage, contributes to flood prevention, travel, navigation, aquaculture, climate adjustment, etc., which have been critical for local economic and social development during and beyond the 1970 s. Most of all, it provides an important water source for around 6 million residents in the Kunming City. In a semi-enclosed state for a long time, DC is faced with increasingly eutrophication and heavy metal pollution due to sewage input, which is not sufficiently exchangeable with clean water in time, and these environmental issues have become more severe [22,23,24]. The deterioration of the water quality has pervasively threatened the water quality of people living around the lake in recent years [23,24]. In order to explore the inherent mechanisms of removal or fixation, research in the aspect of nutrients and organic matter intermediating trace metals’ mobility during their biogeochemical cycles could be an effective approach, but it is scarcely reported.

### 2.2. Sampling and Analysis

Surface water was sampled from the northern site, DC-1, to the southern site, DC-10, in the central water area, and a further three inlet sites I-1, I-2, I-3 and one outlet site of DC-O, were also sampled (Figure 1). Underlying water was restricted to a sample in the central area from DC-1 to DC-10 sites. Sediments were sampled with the sediment column sampler to extract the solid core at DC-2 and DC-9 sampling sites in the central lake area (Figure 1). Sediments were collected with a sediment cylindrical sampler to extract the solid core at DC-2 of the northern lake and DC-9 of the southern lake [24]. Sediments were sliced at 3-cm intervals above 9-cm sedimentation and every 5-cm as the slicing unit from 9-cm downward to 49-cm. A series of sediment samples were centrifuged at 4000 rpm to obtain pore water.

Clean procedures were employed to avoid contamination, all polyethylene sample bottles were acid washed (10% HNO_3_ for 24 h) and rinsed with distilled water four times. Unfiltered water samples were analyzed for hydrological parameters as temperature, pH, electrical conductivity (EC). Total dissolved solids (TDS) and dissolved oxygen (DO) were also recorded with a portable meter (EXO1 Sonde, YSI company) on site. The parameters in instrument were calibrated with standard solutions before taking measurements. Lake water samples for geochemical analysis were filtered in anaerobic conditions using filtering settings, and pore water was filtered using a Millipore Sterivex syringe capsules containing 0.45 μm cellulose acetate filters, then were taken in the plastic bottles, and samples were treated immediately after arrival at the laboratory to avoid potential alterations. Samples were further tested for dissolved organic carbon (DOC), anion and cation analysis, as well as dissolved metals detection in the laboratory.

Slope ratio of S_275–295_/S_350–400_ (S_R_) has been shown to be sensitive to characterizing chromophore dissolved organic matter (CDOM) in natural water, with lower values indicative of DOM higher molecular weight, greater aromaticity and vascular plant input [17,24]. UV-Vis absorbance spectra (200–600 nm, slit 0.5 nm) were recorded using a double beam UV-T9_CS_ spectrophotometer (T9, Persee Company, Beijing, China) on 1-cm quartz cuvette, the ultrapure water being used as reference solution. Fluorescent intensity was detected using a fluorescent instrument, the Hitachi F-7000 (Hitachi Company, Tokyo, Japan). Determination of DOC was performed with a total organic carbon (TOC) analyzer (Analytical Aurora 1030, OI Company). Cations Ca^2+^, Mg^2+^, Na^+^ and K^+^ were measured using an inductively coupled plasma optical emission spectrometer (ICP-OES, Agilent Company, California, USA) and anions (Cl^−^, PO_4_^3−^ and SO_4_^2−^) measurements were performed on a Dionex ICS-90 system (Thermo Scientific Company, California, USA), nutrients (NO_2_^−^, NO_3_^−^ and NH_4_^+^) measurements were performed on Skarlar San++ continuous flow online analyzer (Skarlar Company, Breda, Netherlands), trace metals were determined by ICP-MS (Perkin Elmer, Elan 9000). Quality control of metal determination was assessed with standard references, blank and duplicate samples, and reproducibility of the analytical data was within 5% and the analytical error was estimated to be ≤10%. The Pearson correlation coefficient and the one-way ANOVA were performed using statistical package software (SPSS, version 19).

## 3. Results and Discussion

### 3.1. Characterization of Dissolved Heavy Metal in Lake Dianchi

Temperature in pore water was on average 17.3 °C, which was 2.8 °C higher than that in surface and underlying waters. Water above deposition layer was weakly basic with pH ranging from 7.12–9.07, while there was a narrower range of 7.24–8.48 in pore water [24]. The milliequivalent percentages of Ca^2+^ and Na^+^ in cations were more than 61%, especially at inlets that the proportion was 74.3%, while the proportion values of Mg^2+^ and K^+^ were lower. Nitrate concentrations were high, ranging from 0.02 to 0.72 mmol·L^−1^ in samples of surface and underlying water, and 0.02–1.29 mmol·L^−1^ in pore water. Bicarbonates (HCO_3_^−^) were determined restrictively in water above the deposition layer, and concentration ranged from 1.43 to 3.62 mmol·L^−1^. The molar ratios of Mg^2+^/Ca^2+^ averaged 1.02 except for the inlets, where the mean value was only 0.41. Molar ratios of Na^+^/Cl^−^ averaged 1.03. The milliequivalent ratio of (Ca^2+^ + Mg^2+^)/(HCO_3_^−^ + SO_4_^2−^) was an average of 0.96. After hydrochemical analysis, the water of DC was defined as HCO_3_-Ca type, and the highest HCO_3_^−^ concentration was observed at lake inlet I-3. Water chemistry in Lake Dianchi was dominated by carbonate mineral weathering due to major cation and anions [25,26]. The hydrochemical parameters data have been exhibited in Table 1.

Fourteen major or trace metals were detected, listed as Al, Sc, Ti, V, Cr, Mn, Fe, Co, Ni, Cu, Zn, As, Se, Sr, Mo, Sb, Ba, Pb and U. As illustrated in Table 2, the total amount of metal found was 10.2 × 10^3^ and 19.3 × 10^3^ nmol·L^−1^ in water above the deposition layer and pore water. Here, Ti, Fe, Sr and Ba in water above the deposition layer were considered the main types with concentrations of 3076, 2286, 1645 and 1493 nmol·L^−1^; Al and Zn took second place with 903.8 and 627.3 nmol·L^−1^. Whereas Al, Ti, Fe and Sr in the pore water were in the majority with 5043, 5108, 3813 and 2158 nmol·L^−1^ and Mn, Zn and Ba took second place with 1142, 1248 and 497.0 nmol·L^−1^. In surface water, the total metal concentration was 10.5 × 10^3^ nmol·L^−1^. Fe was detected higher with 4219.5 nmol·L^−1^ at inlets than that at 1934.6 nmol·L^−1^ in the central lake region. Similar to Fe, the highest Ti concentration was also detected to be much more abundant at inlets with an average of 6005 nmol·L^−1^ and up to 6850 nmol·L^−1^ at I-3, whereas only 2628 nmol·L^−1^ than pertained in the central lake region and outlet. Mn ranged from 3.79 to 26.3 nmol·L^−1^, averaging 8.48 nmol·L^−1^ in the surface water of DC. It was 12.5 nmol·L^−1^ on average at inlets, higher than the level of 7.38 nmol·L^−1^ in the central lake region. Al, Co, Cu and Se amounted to highest levels under anaerobic circumstance at inlets, with 1403, 3.93, 20.03 and 7.83 nmol·L^−1^. The total metals in underlying water was 9740 nmol·L^−1^, whereas Ti, Fe, Sr, Ba, Zn and Al were still the most abundant type with average amounts of 2689, 2093, 1567, 1784, 488.4 and 952.5 nmol·L^−1^, respectively. 

The total amount of metals in pore water at DC-2 was 18.3 × 10^3^ nmol·L^−1^, which was 8596 nmol·L^−1^ higher than that of the underlying water. Al, Ti, Fe, Sr, Ba, Mn and Zn were still the predominant type and averaged 5035, 4710, 3799, 2152, 500.7, 987.7 and 888.6 nmol·L^−1^, respectively. The trace metals in pore water at DC-9 were even about 2.0 times that in surface or underlying water. Al, Ti, Fe, Sr, Ba, Mn and Zn were still recognized as the primary metallic components by 5050, 5470, 3826, 2162, 493.7, 1283 and 1575 nmol·L^−1^, respectively. Pore water generally enriched trace elements compared with those in the surface water due to relative long reaction time between water and sediment mineral [1,4]. The average proportion of metals in DOC in pore waters amounted up to 5230.98 nmol·(mg DOC)^−1^, whereas this value was only 2133 nmol·(mg DOC)^−1^ in water above the deposition layer. As the absolute concentrations of metals, the relatively stronger complex for metals still resided in Al, Ti, Fe, Zn, Sr and Ba, the average affinity, and the highest complex value appeared at inlets that averaged 334.2, 2225, 1632, 380.2, 834.3 and 329.8 nmol·(mg DOC)^−1^ for the above mentioned six metals. Generally, the complex value in pore water were 1460, 1522, 1080, 451.2, 593.4 and 146.4 nmol·(mg DOC)^−1^ for Al, Ti, Fe, Zn, Sr and Ba, and was higher than in water above deposition layer (Table 2). Affinity to DOC was characterized by proportions of metals in DOC, which is in agreement with previous studies [18,20].

### 3.2. Analysis of Metal Bound and Liberated Mechanisms

#### 3.2.1. Surface and Underlying Water

The concentrations of metals ranked as Ti > Fe > Sr > Ba > Al > Zn in surface water of DC. In Northern Spain, metals of dissolved forms in a forest watershed were ranked as Fe > Zn > Mn > Pb > Cu > Ni > Cr during flood events, mainly due to abundant Fe and Mn in the Earth’s crust being liberated under reducing conditions [27]. Changes of hydrodynamic environment were very important in determining toxic elements’ distribution and partitioning between the particulate and dissolved phases [28]. In contrast to Fe and Mn, V was fixed at euxinic conditions in the water column or at the sediment/water interface during deposition, which acted as the reductant [29]. Precisely for this reason, dissolved V in surface water was 17.5 nmol·L^−1^ higher than that in pore water in DC. Rivers’ inflow to DC significantly influenced the concentrations of trace metals, e.g., Al, Fe, Sr, Sc, Ti, V, Ni and Zn, as all were much higher at lake inlets than that in the central region and outlet, especially Fe, Ti, Zn and Sr, which were higher at inlets (Figure 2) and the total quantity difference of them four comparing inlets with outlet amounted to 6987 nmol·L^−1^. This result showed that large amount of metals was detained in DC annually. Conversely, Cu, Mo, Sb, Pb, Ba and U had lower concentrations at inlets than in the lake’s central region and outlet, and Ba especially caused the most significant decrease of 406.5 nmol·L^−1^. Thus, the source of metals in surface water might derive from allochthonous input, especially the inflowing rivers around the lake.

In a fluvial system, transport of metals is controlled by a variety of geochemical factors during a series of processes, including mineral weathering, pH, amount and characteristics of both DOC and suspended particulates matter (SPM), redox cycling, precipitation/dissolution and adsorption/desorption reactions [30]. Almost all anions consisted of HCO_3_^−^ and Cl^−^, which might be due to bedrock and minerals weathering around the lake drainage area. The milligram equivalent ratio (Ca^2+^+Mg^2+^)/(HCO_3_^−^+SO_4_^2−^) was close to 1, indicated that the dissolution of carbonate and sulfate minerals determined the hydrochemical composition of DC water. The milligram equivalent ratio of Mg^2+^/Ca^2+^ also reflected the lithology of water erosion area. If this value is from 0.01 to 0.26, suggested a limestone source, and the lithology would be dolomite if this value is above 0.85 [25,26]. The lithology in central lake region predominated with dolomite, but predominated with dolomite limestone at inlets. Na^+^ and Cl^−^ were from halite dissolution. Higher water alkalinity and HCO_3_^−^ at inlets resulted from the anaerobic decomposition of particulate organic matter, and then [H^+^] increased accordingly, which made pH values at inlets lower than in the central lake region. Higher molecular weight and aromaticity of DOM was preferentially fractionated onto iron minerals, and cations (Ca, Na and Cu) intercalated between organic matter and iron by bridging to form organic mineral complexes [31,32]. When oxygen is rapidly depleted, microorganisms use oxidized components such as NO_3_^−^, Mn(IV), Fe(III) and SO_4_^2−^ as electron acceptors for the decomposition of organic matter [33]. SO_4_^2−^ played a critical role as electron acceptor [34] during particulates’ degradation and unbound Ca^2+^, Mg^2+^ in certain ratios (SO_4_^2−^ and ratio of Ca/Mg, R^2^ = 0.75, *p* ≤ 0.01).

Ca/Mg values were used to indicate the paleo-climatic changes and the dynamics of material elements from limestone dissolution [35]. Molar ratios of Ca/Mg averaged 2.45 at inlets and achieved a peak value of 2.54 at I-2, indicated the preferential dissolution of Ca from Mg-calcite and dolomite or from another fresh mineral surface [24,35]. However, this value fluctuated around 1.00 suggesting that the congruent dissolution for Ca and Mg from limestone resulted in the linear reaction path in the central lake region. Oxidation of DOM was accompanied by a reduction of SO_4_^2−^, both of which were consumed dramatically, especially at lake inlets (SO_4_^2−^ and DOC, R^2^ = 0.98, *p* ≤ 0.01), which indicated that gathering of sulfate-reducing bacteria contributed greatly to DOM decomposition [33,36]. Metals were liberated during the aforementioned degrading processes of particulate organic matter (total metals and the ratio of Ca/Mg and Sr/Mg, R^2^ = 0.83, *p* ≤ 0.01) and SO_4_^2−^ was also utilized as electron acceptor, and this part of particulate organic carbon (POC) was defined as mineral-associated organic carbon [37,38]. Sr was recognized to be released from congruent dissolution of Mg-calcite and strontium probably represents a minor impurity in calcite crystal lattice [7,15]. In this research, Sr/Mg had a similar horizontal trend to Ca/Mg although it was three orders of magnitude lower than Ca/Mg (Ca/Mg and Sr/Mg, R^2^ = 0.92, *p* ≤ 0.01), indicating that Sr, as a minor impurity in limestone, was dissolved out along with dissolution of Ca rather than congruent dissolution with Mg. Molar ratio of 4.01 × 10^−3^ for Sr/Mg at inlets was higher than that of 2.42 × 10^−3^ in the central lake region, indicating [Sr] enrichment probably associated with different dynamics of Sr and Mg leaching; this might be caused by a substantial release of Mg from dissolving dolomite compared to the decreasing Sr release from Mg-calcite [35]. Land runoff brought in abundant metallic elements and minerals through river input, Ca^2+^, Mg^2+^, Na^+^ and K^+^ amounted to 4.89 mmol·L^−1^ and total trace metals reached up to 17.2 × 10^3^ nmol·L^−1^ at 3-I.

Dissolved Fe was much higher at inlets than the central lake region due to reduction of Fe(oxy)hydroxide under relatively anaerobic conditions [39,40] and it fluctuated around 1935 nmol·L^−1^ in the central lake region and outlet. It was pointed out that Sr, Fe and Mn were contained in calcite structure, clay minerals and pyrite [35]. Dissolved Fe was one of the metallic elements that originated from limestone disintegration (Fe and ratios of Ca/Mg, Sr/Mg, R^2^ = 0.96, *p* ≤ 0.01). Ti, Sc, V, Cr, Zn, Fe, Ni and Sr were similar to Fe (R^2^ = 0.99, *p* ≤ 0.01), exhibiting a decreasing horizontal trend from the direction of inlet to outlet and originating from particulate organic matter (POM)/SPM decomposition (Figure 2). Similar to Fe, the highest Ti concentration was also detected to be much more abundant at inlets than that pertained in the central lake region and outlet.

Sc, Ti, Cr, Fe, Ni and Sr were anti-correlated with S_R_ (R^2^ = 0.80, *p* ≤ 0.01), indicating that molecular size impacted their horizontal distribution. The higher molecular weight of DOM carried relatively more metals at lake inlets, as Zn, V, Mn, Se and Co. Dissolved U dropped to a minimum as DOC did at inlets where the minimum values were 0.94 nmol·L^−1^ and 0.26 mmol·L^−1^, and both varied consistently in horizontal distribution (R^2^ = 0.66, *p* ≤ 0.01), whereas molecular weight was relatively higher at inlets than that in the central lake area (U and S_R_, R^2^ = 0.90, *p* ≤ 0.01). It was postulated that small molecules aggregated into large size to complex U and adsorbed/co-precipitated onto minerals surface, then colloidal DOM was dragged into POM. This condensation polymerization had a great influence on the U distribution. Another plausible explanation could be that when redox potential is sufficiently low, U(VI) can be reduced into insolubility and be removed from the water column [41]. When dissolved oxygen was comparatively deficient and only 0.17 mmol·L^−1^ at inlets, dissolved U was also the minimum at inlets. Alternatively, organic matter could play an important role as a reductant that resulted in a change from U(VI) complex to inorganic U(IV) in terms of the U-bearing minerals within the cells, fractures and veinlets in inertinite macerals [42]. Dissolved Mo and Sb performed similarly with U and showed the same distribution patterns with DOC, SO_4_^2−^ (R^2^ = 0.82, *p* ≤ 0.01), which were all related to microbial decomposition of organic matter. It was suggested that EPS (extracellular polymeric substances) chains can intercalate into montmorillonite layers by hydrogen bonding connection and trace metal added a bridge between EPS and montmorillonite [38,43]. Thus, breakage of this bridged bond would also result in the dissolution of trace metal. It was observed that Mn and DO showed totally converse trends in surface water (R^2^ = 0.62, *p* ≤ 0.01), demonstrating Mn distribution was obviously affected by change of oxygenation and dissolved Mn originated from reduction of manganite [5]. Fe and Mn play an extremely important role in estuarine geochemistry because their oxides/hydroxides and sulfides are effective scavengers for a wide range of trace metals [16,44]. Once redox conditions changed, Al, Co, Cu and Se would be repartitioned between particulates and water and their distributions corresponded with Mn (Figure 2). Oxygen deficiency at inlets should be responsible for release of these metals during Fe or Mn reduction.

Cations and anions in underlying water could be divided into two types of salts, which are mutually anti-correlated (R^2^ = 0.74, *p* ≤ 0.01); one was carbonates represented by HCO_3_^−^, NO_3_^−^, Ca^2+^ and Mg^2+^, the other was halides represented by SO_4_^2−^, Cl^−^, K^+^ and Na^+^. The ratio of these two mineral salts [carbonates]/[halides] was relatively significant at inlets especially 1.49 at 1-U and then fluctuated around 1.04 in other sites of central lake region. The above results indicated that ion exchange occurred between the two salts and then constituents and structure were re-arranged in lime dolostone, and this lattice constituents and structure changed little in dolomite area maintained stable. Ti, Fe and Sr together with Co and as had a consistent horizontal distribution of a decreasing trend in the direction of water flow, which varied according to ratios of Sr/Mg and Ca/Mg (Figure 3), indicating that these metals were liberated from the related lattice of carbonates. Dissolved Ba and Ni showed positive and negative correlations to molar ratios of Na/K, closely related to mineralogical structure and mineralization of particulate halides, which impacted migration and partition of Ba and Ni between waters and particles (Figure 3).

#### 3.2.2. Pore Water at DC-2 and DC-9

In pore water at DC-2, SiO_3_^2−^, NO_2_^−^ and PO_4_^3−^ were observed highest above the 9-cm sedimentation layer, at 214.1, 11.9 and 6.96 μmol·L^−1^, respectively. Microbes were observed to bind sediment using extracellular polysaccharides and occurred in patchy distributions within sediments [43]. DOC formation was usually accompanied by nutrient mobility in pore water during early diagenetic reactions. Specific ultraviolet absorbance (SUVA) of SUVA_254_ and SUVA_260_ indicated aromaticity and the hydrophobicity of organic matter [24]. E_253/203_ referred to substitutable sites on aromatic rings of organic matter and substituent species. The optical characteristics suggested that DOC mainly originated from authigenic micro-organic residues and algal excrete/detritus in the pore water.

Generally, dissolved oxygen penetrated into sediments for a few centimeters on average, causing partial DOC oxidation [1,24]. Available oxidants became too limited to be utilized below this depth, nitrate and Fe/Mn oxides’ abundance decreased dramatically, and Fe- and Mn- oxides or hydroxides disintegrated rapidly in an anaerobic environment [45]. Mn(IV) together with PO_4_^3−^ and SiO_3_^2−^ were utilized as electronic acceptors, rapidly reduced to Mn(II), resulting in the peak value 4111 nmol·L^−1^ at 6-cm. Thereafter, dissolved Mn abruptly dropped to 1106 nmol·L^−1^ at 19-cm and further decreased to only 20.2 nmol·L^−1^ at 24-cm. Sc, Co and as bound closely with Mn-bearing minerals, and their dissolved forms also decreased with depth. Peak values of them were 62. 5, 12.4 and 38.7 nmol·L^−1^ above 14-cm depth (Figure 2). Molar ratios of Ca/Mg varied conformably with pH values (R^2^ = 0.52, *p* ≤ 0.01) and both increased with depth. This indicated that pore water was increasingly alkaline with disruption or hydrolysis of carbonatite. Along with the change of mineral lattice in depth, Al, V, Cr, Ni, Cu, Zn, Se, Mo, Sb, Ba and U were more dissolved out from limestone phase with alkalinity and ratio of Ca/Mg increasing in depth (Figure 4). Trace metals are also likely to be scavenged or removed by sediments because of adsorption, co-precipitation or complexation from dissolved phase in aquatic ecosystems [38,46,47]. It was found that Sc, As and Sr were predominantly regulated by carbonatite and the hydrated oxide lattice structure (Sc, As and Sr anti-correlated to ratio Ca/Mg, R^2^ = 0.74, *p* ≤ 0.01; Figure 4), demonstrating that Sc, As and Sr were reversely complexed with or wrapped in solid inclusions when Ca^2+^ or Mg^2+^ leached out with disruption, or perhaps cations exchange happened [48,49].

Trace metals were closely correlated with the clay mineral phase, and were influenced critically by microbes and referred to P, S, C and N cycling [38,50]. SO_4_^2−^ and NH_4_^+^ in pore water at DC-9 contributed to DOC production (R^2^ = 0.42, *p* ≤ 0.05), due to corresponding microbial activities that resulted in a decrease for nutritive elements and an increase for DOC. Slightly different from the results in pore water at DC-2, the positive correlations between total metal level (Zn, Fe, Ti, V, Cr and Cu), especially Fe in pore-water at DC-9 and hydrochemical parameters (pH values and ratios of Ca/Mg), were more significant (R^2^ = 0.75, *p* ≤ 0.01; Figure 5). Fe primarily originated from POM, which bound to the Fe-bearing mineral phase (Fe and DOC, R^2^ = 0.55, *p* ≤ 0.01), indicating that Fe was probably incorporated into Ca-, Mg- mineralogical lattice of clay as carbonates, organo-mineral complexes, silicates, oxides, oxyhydroxides, etc. The fixation and liberation of metals were closely related to crystallization and hydrolysis of minerals. It was reported that microbially mediated reduction reactions of Fe(OH)_3_, SO_4_^2−^ and NO_3_^−^ consumed protons when organic matter oxidized and were responsible for the increase of pH [33]. Thus, reduced Fe(II), together with certain ratios of Ca/Mg, were unbound from clay carbonate [31,32]. Co, Sc and Sr were observed to decrease obviously with depth, which was quite the opposite of the ratio of Ca/Mg (R^2^ = 0.7, *p* ≤ 0.01), having been explained as the same mechanisms to pore water at DC-2 that reverse complexation/encapsulation or cations exchange happened [48,49]. As illustrated in Figure 5, the sum amount [M2] of Sc, Co and Sr was anti-correlated with [M1] (R^2^ = 0.75, *p* ≤ 0.01) ([M1] denoted the total amount of U, Mo, Zn, Fe, Ti, V, Cu and Cr), the explanation of cation exchange was further supported. Ti, V, Cr, Cu, Zn, Mo and U performed profiles similar to Fe, especially Ti and Zn (R^2^ = 0.9, *p* ≤ 0.01). The oxidized state of Mn(IV) or Mn (III) was abruptly reduced to Mn(II) and dissolved out greatly at 14-cm depth, and the dissolution quantity amounted to 12.6 × 10^3^ nmol·L^−1^. Ni performed consistently to Mn and unexpectedly rose to 94.15 nmol·L^−1^ at 14-cm (Figure 5), and in addition, did not change obviously.

The average content of metals in water above the deposition layer was lower than that in pore water. High level of total metals in water above the deposition layer decreased along the direction of water flow, which indicated the significant influence of natural biogeochemical processes on metal quantities in the aquatic eco-environment and the attendant potential hazard of anthropogenic emissions on water quality security. By contrast, metals concentrations increased in pore water along the sediment core downward, demonstrating that metals gradually preserved with sediment aged, and would be released once the deposition condition changed as the second pollution source. The addition or depletion trends of metals’ spatial distribution in natural waters were influenced by hydrochemical conditions (redox conditions, pH, nutrients, etc.), clay components and structure, cation exchange and organic complexation/adsorption [38]. Dissolved Fe and Mn were unbound in reducted forms due to oxygen deficiency. U was dragged to particles by bio-molecules aggregation. However, mineral-elements Fe and other metals were liberated through a series of redox reactions in which oxidized Fe(III) and POM anticipated, carbonate ingredients Ca and Mg were unbonded with certain rations. Mn was mainly utilized as electric acceptors under oxygen deficiency during diagenesis, having been reduced to Mn(II) and dissolved out into water.

### 3.3. Properties of Metals Affinity to DOM

In surface water, the proportion of total metals in DOC averaged 2491 nmol·(mg DOC)^−1^. The proportions of metals presenting in DOC mainly depended on the amount of dissolved metals; a higher proportion of 5818 nmol·(mg DOC)^−^^1^ was at inlets, whereas 1584 nmol·(mg DOC)^−1^ was present in the central lake region. As for Fe, the ratio in DOC was 1632 nmol·(mg DOC)^−1^ at inlets and higher than that at 341.0 nmol·(mg DOC)^−1^ in the central lake region. [M]/DOC was 1776 nmol·(mg DOC)^−1^ in underlying water. [M]/DOC was nmol·(mg DOC)^−1^ in pore water at DC-2. Ti, Fe and Sr as the primary species occupied mean proportions of 1232, 1013 and 568.4 nmol·(mg DOC)^−1^, respectively. Ratio of [M]/DOC was highest in pore waters at DC-9 among the four water compartments and the average value was 5600 nmol·(mg DOC)^−1^. Proportions of total metals in DOC in pore water was 2.5 times that in water above the deposition layer, with average values of 5231 and 2133 nmol·(mg DOC)^−^^1^, respectively. Total proportions of metals in DOC at site DC-2 increased in a zigzag with depth; the minimum value was 1235 nmol·(mg DOC)^−1^ at 29-cm and the maximum was 7530 nmol·(mg DOC)^−1^ at bottom sediment 44-cm. Whereas proportions changed in an arc shape at DC-9, the maximum value was 8668 nmol·(mg DOC)^−1^ at 14-cm. Drying-wetting cycles impacted much on mobilization potential of Cu (CuMP) and caused smaller CuMP of 9 nmol·(mg DOC)^−1^ than 17 nmol·(mg DOC)^−^^1^ when no drying-wetting cycles applied [51]. Organic matter with the metal-binding sites in the sediment could be the primary metal-binding ligands for trace metals, suggesting significant in metal coordination by DOM [19,20]. In comparison, Cu in this study has a much lower MP of 2.66 nmol·(mg DOC)^−1^ in the water above deposition layer, and an equivalent CuMP of 8.63 nmol·(mg DOC)^−1^ in pore water to that value under drying-wetting cycles. This result is related to the higher abundance of Cu and smaller abundance of DOC in pore water.

DOC is an important factor that controls partition of elements in solid and solution through a series of complicated biogeochemical reactions, which reflect their relationships and the spatiotemporal variation of these elements, e.g., dissolved trace metals (e.g., Cu, Zn and Cd) in natural aqueous systems [5,7,20]. The optical properties of DOM could be useful to understand the trace metal-DOM interactions and metal bioavailability in natural and effluent systems [20]. SUVA_254_ has been indicating the number of unsaturated C–C conjugated double bond and has been used as a surrogate measurement for DOC aromaticity [52]. SUVA_254_ is an important DOM qualitative factor that determines concentrations of trace metal complexes in natural and effluent waters, and trace metals are preferentially complexed by the metal binding ligands (e.g., acidic functional groups) present in the proximity of aromatic structures of humic materials, which determined the metal binding affinity [20,53]. Aromatic structures were commonly found in hydrophobic components in the waters of Dianchi Lake, which could be discerned from the significant correlations between SUVA_254_ and SUVA_260_ (R^2^ = 0.99, *p* ≤ 0.01). Although the DOM in DC originated from an authigenic source, it was still specifically different in terms of molecular structure and size in these four water compartments.

#### 3.3.1. Surface and Underlying Waters

DOM is a mixture of diverse components, and metal-binding affinity is defined by the concentration ratio of DOM-bound metals relative to DOC. Cu and Fe in natural waters and effluents are significantly influenced not only by DOM concentration but also by the aromaticity of DOM [20,26]. The assessment of DOM optical properties may be useful to understand the trace metal-DOM interaction and metal bioavailability in natural and effluent water systems [10]. In surface water, the proportion of total metals in DOC mainly depended on the amount of dissolved metals and was intimately related to molecular size and the percentage of substitutable aromatic rings in DOM (R^2^ = 0.64, *p* ≤ 0.01; Figure 6). Sc, Ti, Cr, Fe, Ni and Sr were anti-correlated with S_R_ and E_253/203_ (R^2^ = 0.8, *p* ≤ 0.01), indicating that molecular size and the percentage of substitutable groups on aromatic rings impacted their horizontal distribution. When E_253/203_ values rose, the substitutable groups of different number and species on aromatic rings should increase accordingly. DOM of a higher molecular weight (corresponded to a lower S_R_ value) carried more metals at lake inlets, such as Zn, V, Mn, Se, Cu, Mo and Co in addition to Sc, Ti, Cr, Fe, Ni and Sr (Figure 6). In contrast, the more species of substitutable groups as –COOH, –OH, HCO–, –CO–, etc. whereas coincided with the reducing proportions in the central lake region. This resulted from the much more concentrations of metals at inlets than lake central region. Taking Ti and Fe as an example, their proportions levels at inlets were more than two times of that in central lake region. Dissolved Mn, Se and Mo mainly complexed with protein-like components in DOM ([M]/DOC and [FI_p_]/DOC, R^2^ = 0.95, *p* ≤ 0.01) ([FI_p_]: fluorescent intensity of protein-like components; [FI_p_]/DOC: proportions of protein-like in DOM).

In underlying waters, the proportion of Mn in DOC was regulated by the molecular weight of DOM, which was highest of 0.05 × 10^−3^ at site DC-6 and S_R_ value also rose up to a maximum of 2.30 ([Mn]/DOC and S_R_, R^2^ = 0.74, *p* ≤ 0.01), indicating that Mn mainly complexed with DOM of low molecular weight. The combination of some trace metals with DOM depended on structures of DOM, higher aromaticity and hydrophobicity caused firmer complexation to trace metals [17,20]. Proportions of Sc, V, Co, Cu, Ni, As, Mo, U and Se in DOC showed consistent variation trends with SUVA_254_ and SUVA_260_ values (Figure 7). In fact, dissolved Sc, V, Mo and U bound with protein-like substances ([M]/DOC and [FI_P_]/DOC, R^2^ = 0.79, *p* ≤ 0.01), and Se mainly associated with humic-like substances ([M]/DOC and [FI_H_]/DOC, R^2^ = 0.7, *p* ≤ 0.01; [FI_H_]: fluorescent intensity of humic-like components; [FI_H_]/DOC: proportions of humic-like in DOM). It is reported that the abundance of PAHs (polycyclic aromatic hydrocarbons) is in accord with the complex formation potential of the size fractionated DOM with inorganic species (metals) [54]. In fact, complex of aromatic rings with metals depended on substitutable groups of diversified species and number. Non-substitutable groups, such as straight-chain alkane and their strong substitution on aromatic rings, hindered complexation for metals to DOM. When values of E_253/203_ decreased, the number of substitutable groups (oxhydryl, carboxyl, carbonyl and ester group) and species on aromatic rings decreased and straight-chain alkane increased, which can hardly be substituted [55]. Combination of Ti, Fe and Sr to DOM attenuated with E_253/203_ increased (Figure 7), mainly due to a significant decrease of metal levels. Ti, Fe and Sr showed the decreasing quotients in DOC along the direction of water flow. Affinity of Fe to DOM was also promoted by aromaticity and hydrophobicity apart from Fe abundance and performed consistently with SUVA values (Figure 7).

The complexation between metals and DOM was closely related to structure, size and abundance of metals, and the most abundant organo-metals appeared in surface waters at inlets. Obviously, SUVA can still be utilized to indicate the association of metals to DOM although it has a limitation in reflecting the chemical composition of DOM related to its potential complexation with metals [51,54]. In general, Ti, Fe and Sr were principally controlled by content level. Mn bound with different size of molecular fractions in surface and underlying water, with high or low molecular weight DOM, respectively. Various molecular structures and sizes constituted distinctive DOM [17,53], which caused different metal behaviors in two water compartments.

#### 3.3.2. Pore Water

Three types of metals were influenced by different properties of DOM during their combination. Aromaticity and hydrophobicity became the main factors that controlled combinations of Ti and Sr to DOM. Proportions in DOC of Ti and Sr unexceptionally minimized to 4.42 × 10^−3^ and 1.78 × 10^−3^ at the 29-cm sediment depth due to relatively single DOM structure from data of SUVA_254_, SUVA_260_ (Figure 8). The ratios of U and Ba, Mo and Zn to DOC increased with the S_R_ value decreasing in depth (Figure 8), indicating U, Ba, Mo and Zn mainly complexed with large size molecules in DOM ([M]/DOC and S_R_, R^2^ = 0.61, *p* ≤ 0.01). [FI_H_] means fluorescent intensity of humic-like components. There is good relationship between [Ba]/DOC) and [FI_H_]/DOC (R^2^ = 0.54, *p* ≤ 0.01). There, Ba could precisely form complexes with humic-like components, the combined quantity in DOC for Ba had a similar profile to the ratio of humic-like in DOC. The third combination of metals to DOM was closely related to molecular weight, proportions of Co, Mn, As and Sc in DOC positively correlated to S_R_ of DOM, both maximums of total proportion and S_R_ appeared at 6-cm with 15.2 × 10^−3^ and 1.19, here, the ratio of [Mn]/DOC was as high as 14.85 × 10^−3^. It was manifested that dissolved organo-Mn/Co/As/Sc mainly existed in low-molecular weight fragments (Figure 8).

Proportions of Ti, Fe and Sr to DOC in pore water at site DC-9 gradually decreased with depth (Figure 9). As illustrated, contributions to DOC of Sc, Co and Se decreased with the aromaticity and hydrophobicity of DOM attenuated (Figure 9), indicating weaker affinity for DOM to complex metals when DOM was simply constructed. Proportions of [Zn], [Mo] and [U] in DOC gradually increased with decreasing molecular weight of organics along the sediment profile, with highest ratios of 8.23, 0.18 × 10^−3^ and 0.02 × 10^−3^ at depth 19-, 29- and 29-cm, respectively (Figure 9), whereas S_R_ also fluctuated around a range of higher values of 1.03–1.18 in depth of 19–29-cm.

The molecular structure and size of DOM greatly impacted complexation between DOM and metals in the waters of DC, e.g., substitutable structure on aromatic rings, aromaticity and hydrophobicity and molecular weight were all critical factors [17,20,53]. The binding of the same metal to DOM in pore water was discerned to be influenced by different organic properties [11], e.g., Co at DC-2 was mainly regulated by molecular size, whereas it was regulated by aromaticity at DC-9. Complexation of Zn, Mo and U to DOM in pore water at these two sites was regulated by the same factor as molecule size, but they were complexed with different molecular weight fractions. The differentiated combination mechanisms led to the two differential profiles of [TM]/DOC in pore water at DC-2 and DC-9. This discussion highlighted the importance of DOM molecular information in the understanding of DOM-metal interactions. These results were due to discriminating early diagenesis and biological behavior of sediments [11,17,56].

## 4. Conclusions

This study investigated the distribution of trace elements and DOC in the lake waters and pore waters at the Dianchi Lake, SW China. Distribution of trace metals in the water column and pore water in sediment was related with the binding nature between the specific metal and organic/inorganic phases, Fe-, Mn-oxyhydroxides or biogenetic carbonate (calcite and dolomite). Once redox conditions changed, major trace metals would be repartitioned between particulates and water. The mobility of metals was governed by competition between precipitation/sorption and complexation. The characterization and analysis of trace metals and organic matter in an aquatic environment promoted the understanding of material cycles and environmental behavior traits in eutrophic lake ecosystems. The distribution and characteristics of dissolved organic matter greatly impacted complexation between DOM and metals in the waters of DC. Organic properties would be very critical to complex and unbind metals, as well predominate their fate in relevance of corresponding eco-environmental effect in aquatic ecosystems. Finally, it provided valuable references and guidance for policy and decision-making on pollution status. The studied results will be conducive to ecological restoration and environmental protection in DC drainage.

## Figures and Tables

**Figure 1 ijerph-16-04919-f001:**
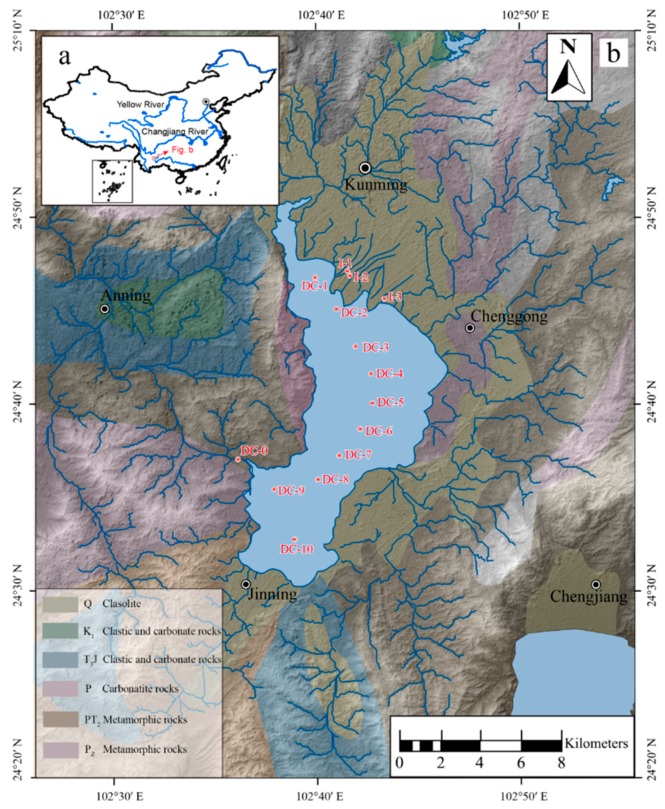
Location map of Dianchi Lake showing sampling sites (I: lake inlet; O: lake outlet).

**Figure 2 ijerph-16-04919-f002:**
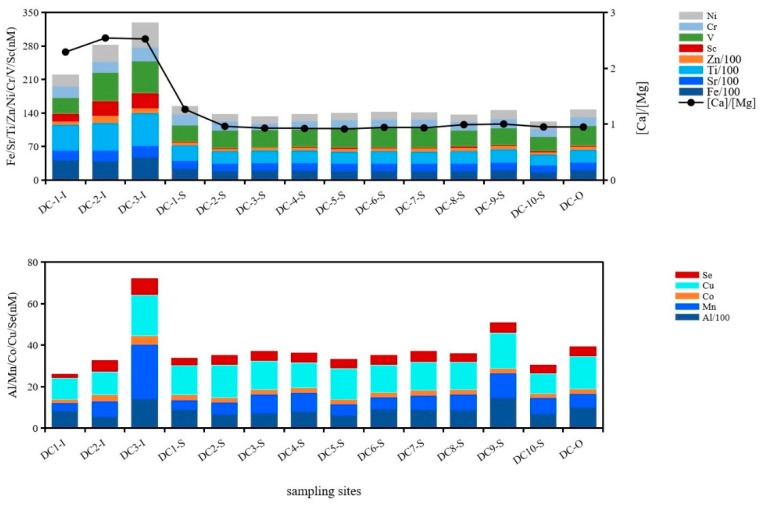
Decreasing horizontal variation trends under influence of dissolved oxygen level on mineral disintegration and metal concentrations in direction of water flow in surface waters of Dianchi Lake.

**Figure 3 ijerph-16-04919-f003:**
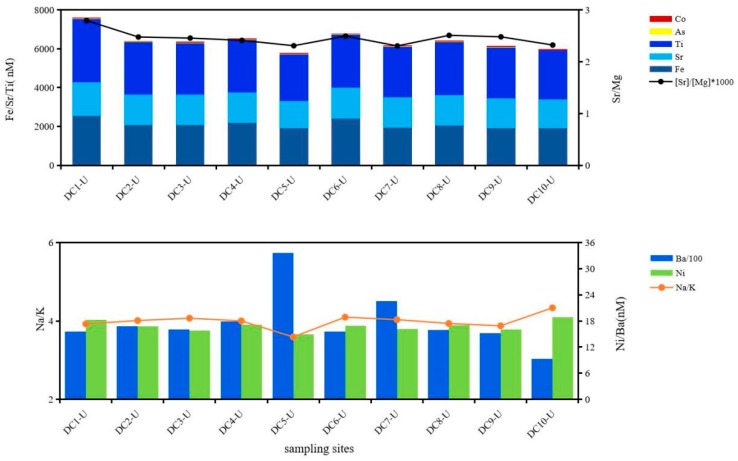
Horizontal variation trends of metals and influence of lattice rearrangement of minerals on metal concentrations in direction of water flow in underlying waters of Dianchi Lake.

**Figure 4 ijerph-16-04919-f004:**
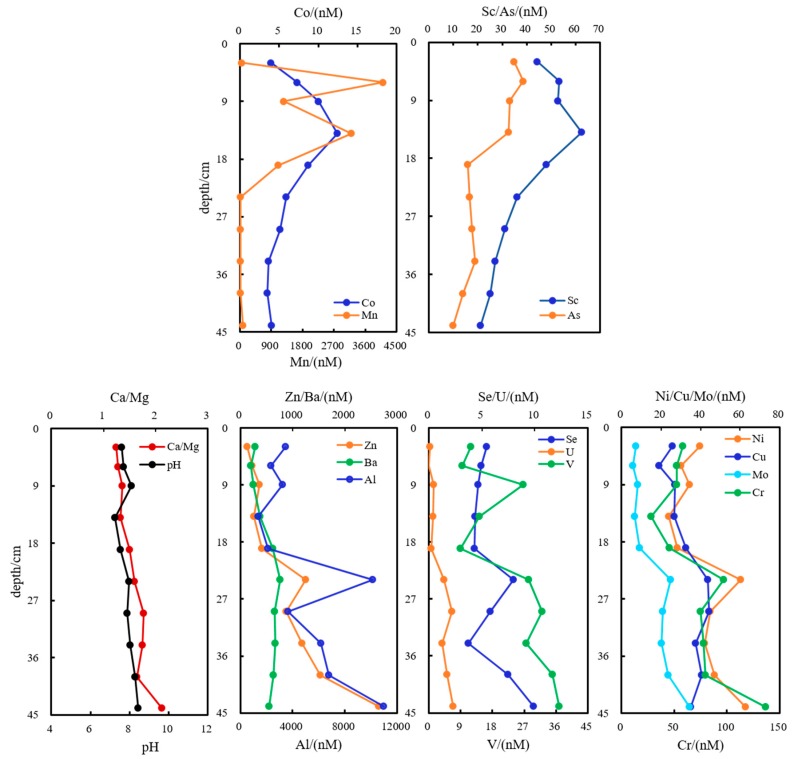
Increasing and decreasing profiles for metals, influence of mineral disintegration on metals concentrations in pore waters at DC-2.

**Figure 5 ijerph-16-04919-f005:**
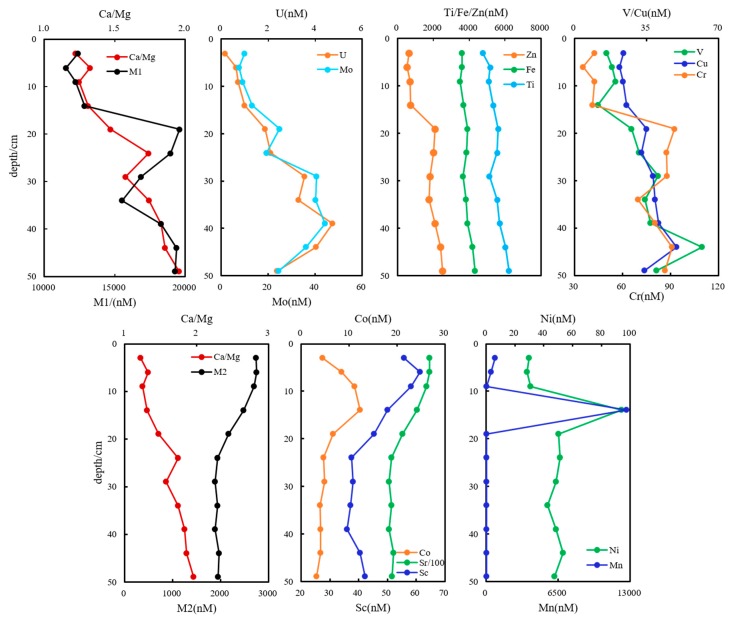
Increasing and decreasing profiles for metals and influence of mineral disintegration and ion exchange on levels of metals in pore waters at DC-9. M1 denoted the total amount of U, Mo, Zn, Fe, Ti, V, Cu and Cr. [M2] denoted the total amount of Sc, Co and Sr.

**Figure 6 ijerph-16-04919-f006:**
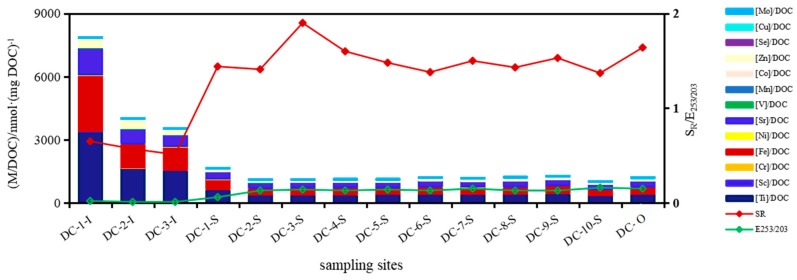
The decreasing affinity for metals binding to dissolved organic matter (DOM) and the influence of molecular sizes and structures on affinities in direction of water flow in surface waters.

**Figure 7 ijerph-16-04919-f007:**
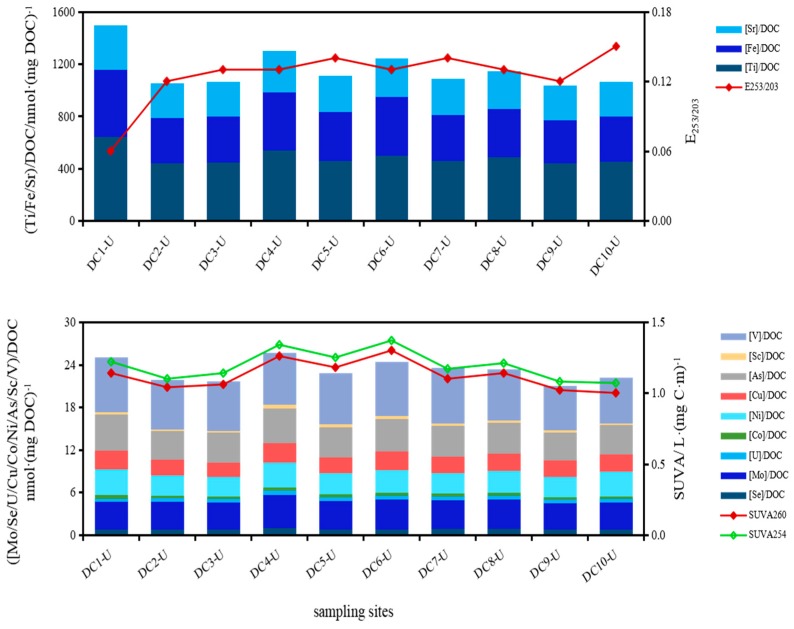
Affinity diversification of metals binding to DOM and the influence of molecular structures on affinities in direction of water flow in underlying waters.

**Figure 8 ijerph-16-04919-f008:**
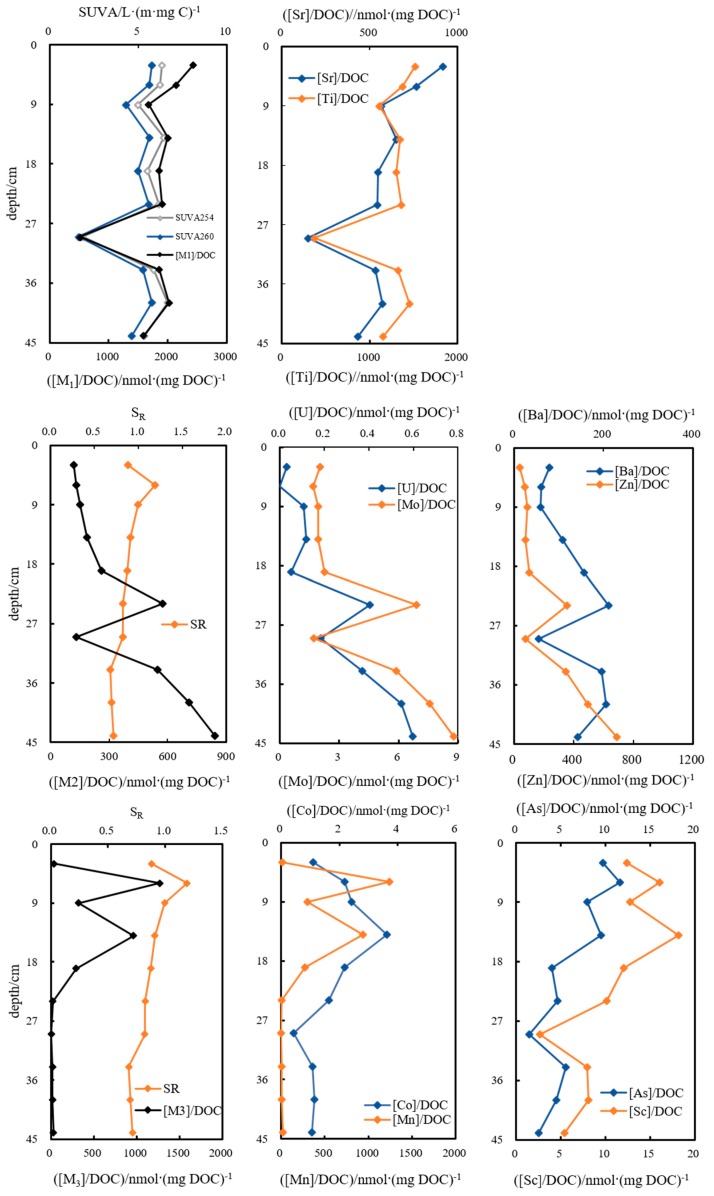
Affinity diversification of metals binding to DOM and the influence of molecular structures and sizes on affinities along the sedimentary profile in pore waters at DC-2.

**Figure 9 ijerph-16-04919-f009:**
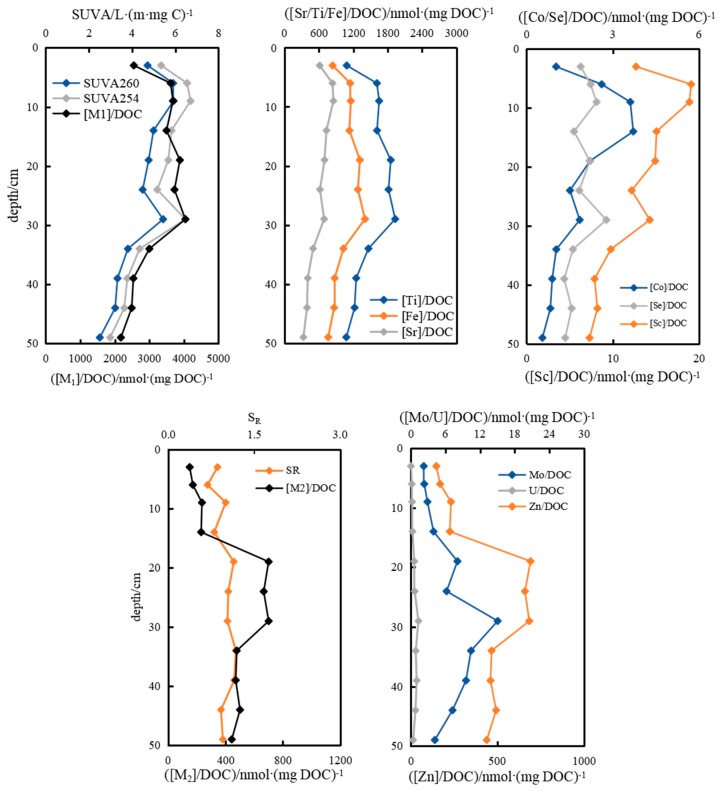
Affinity diversification of metals binding to DOM and the influence of molecular structures and sizes on affinities along the sedimentary profile in pore waters at DC-9.

**Table 1 ijerph-16-04919-t001:** Hydrochemical properties in surface-, overlying- and pore-waters of Dianchi Lake.

Stations	EC (µS·cm^−1^)	TDS (mg·L^−1^)	pH	DO (mg·L^−1^)	Chl (μg·L^−1^)	DOC (mmol·L^−1^)	FI	HCO_3_^−^ (MER)	Cl^−^+NO_3_^−^ (MER)	SO_4_^2−^ (MER)	Ca/Mg Molar Ratio	Na/K Molar Ratio
Surface-	293.8	240.6	8.61	8.14	6.71	5.82	1.89	1.65	1.02	1.02	0.98	3.95
1-I	308.4	239	7.89	7.27	2.33	1.59	1.87	2.94	0.39	0.51	2.29	1.46
2-I	467.2	353	7.12	5.94	1.27	3.43	2.09	2.77	2.07	0.72	2.54	1.04
3-I	586.8	448	7.13	2.98	1.95	4.35	2.07	3.62	2.80	0.85	2.53	1.1
O	304.2	240	9.07	10.51	13	5.98	1.88	1.59	1.03	1.05	0.95	3.97
Overlying-	295.9	242.7	8.72	7.82	13.5	5.55	1.9	1.62	1.02	1.04	0.99	3.99
Pore-2	365.3	—	7.88	—	—	4.43	2.06	—	2.38	0.32	1.56	2.83
Pore-9	374.9	—	8.12	—	—	3.81	2.06	—	2.93	0.34	1.57	2.57

“—”: undetected; Surface-: means for parameters in surface water; I: lake inlet water; O: lake outlet water; overlying-: overlying-water; Pore-2: porewater at No.2 sampling site; Pore-9: porewater at No.9 sampling site. FI: fluorescent index; MER: milliequivalent ratio.

**Table 2 ijerph-16-04919-t002:** Concentrations of metals and their proportions in dissolved organic carbon (DOC) in water compartments of Dianchi Lake.

Stations/Units Concentrations/Proportions			Al	Ti	Fe	Zn	Sr	Ba	V	Cr	Mn	Ni	Cu	As	Mo	Sb
Surface-	Con.	nM	840.0	2633	1965	627.1	1556	1324	36.67	17.67	7.47	16.65	13.88	24.22	22.33	7.01
	Prop.	nM/(mgDOC)^−1^	145.8	456.8	340.5	108.5	269.0	228.5	6.32	3.07	1.28	2.88	2.40	4.19	3.84	1.21
1-I	Con.	nM	822.4	5433	4218	701.2	2022	528.2	33.65	24.32	3.79	25.17	10.37	39.95	7.14	3.24
	Prop.	nM/(mgDOC)^−1^	517.2	3417	2653	441.0	1271	332.2	21.17	15.30	2.39	15.83	6.52	25.13	4.49	2.04
2-I	Con.	nM	557.1	5762	3941	1535	2335	1863	60.37	22.17	7.44	36.15	11.08	27.24	27.37	4.10
	Prop.	nM/(mgDOC)^−1^	162.4	1680	1149	447.4	680.7	543.3	17.60	6.46	2.17	10.54	3.23	7.94	7.98	1.19
3-I	Con.	nM	1404.3	6869	4761	1097	2396	495.3	67.02	28.74	26.35	54.69	19.85	40.64	24.95	4.55
	Prop.	nM/(mgDOC)^−1^	322.8	1579	1095	252.1	550.9	113.9	15.41	6.61	6.06	12.57	4.56	9.34	5.73	1.05
O	Con.	nM	989.9	2657	2070	660.1	1605	1806	42.49	17.39	6.60	16.63	15.72	25.41	23.07	6.45
	Prop.	nM/(mgDOC)^−1^	165.5	444.3	346.1	110.4	268.5	302.0	7.11	2.91	1.10	2.78	2.63	4.25	3.86	1.08
Overlying-	Con.	nM	953.2	2697	2137	492.8	1574	1780	39.82	21.27	8.38	17.17	13.31	24.29	22.46	6.81
	Prop.	nM/(mgDOC)^−1^	171.9	488.3	387.5	89.47	284.6	324.0	7.18	3.83	1.52	3.11	2.40	4.39	4.05	1.23
Pore-2	Con.	nM	5038	4723	3878	896.61	2162	499.5	23.16	70.23	988.8	42.25	32.92	23.23	15.94	0.78
	Prop.	nM/(mgDOC)^−1^	1339	1232	1013	234.09	568.4	127.4	5.86	18.13	280.2	10.91	8.36	6.15	4.07	0.19
Pore-9	Con.	nM	5054	5485	3906	1590	2172	492.5	30.74	68.79	1284	48.88	32.74	26.18	24.26	1.28
	Prop.	nM/(mgDOC)^−1^	1392	1506	1072	424.62	605.3	138.8	8.20	18.88	381.2	13.59	8.90	7.18	6.57	0.34

Surface-: surface water; I: lake inlet water; O: lake outlet water; over-: overlying-water; Pore-2: porewater at No.2 sampling site; Pore-9: porewater at No.9 sampling site. Con.: average concentration; Prop.: proportion of metals in DOC; nM: nmol·L^−1^.

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
