# Peer review of "Spatial Variations of Trace Metals and Their Complexation Behavior with DOM in the Water of Dianchi Lake, China"

_ijerph, 2019, doi:10.3390/ijerph16244919_

Round 1
Reviewer 1 Report
Yi et al. conducted substantial work to measure metal concentrations and [DOC], trying to correlate the distribution of metals with organic matter signals in DC lake. There was no clear conclusion made throughout the paper. Based on the data presented in this manuscript, "complexation behavior" could not be discussed, and should not be included in the title to confuse the readers. The simple discussion made between concentrations of metals and concentrations of DOC, cannot be used to deduce the complexation of metals to DOM. More detailed study of DOM composition and kinetic experiments of the binding between metals and DOC will be necessary to illustrate the complexation of metals with DOM.
Substantial work is needed in order to improve this manuscript. The discussion was not well related with the data presented. Authors cited references to make discussion, while I don't see the authors interpreting their own data to make conclusions. Numbers of places in the manuscript text just illustrate concentrations and correlations of metals and DOC, which were supposed to be presented by tables and figures.
The authors mixed up all possible reasons for the observed parameters, but giving no useful or definite conclusion to the readers. This makes the manuscript not interesting at all.
Reviewer 2 Report
This study characterized the distribution of trace metals and the optical properties of dissolved organic matter (DOM) on the surface, and the underlying and pore water of Dianchi Lake (DC) to understand the origin of metals and complexation mechanisms to DOM. Three issues were discussed: (1) dissolved metal concentrations in DC, (2) analysis of metal bound and liberated mechanisms, and (3) properties of metal-DOM complexation. The metal-DOM complexation is an important issue; however, the following questions need to be addressed.
(1) The definitions, measurements, and calculations of the optical indicators, SR, E253/203, SUVA260, and SUVA254 and fluorescent intensity were not found in Section 2 Material and Methods. Please include the definitions, measurements, and calculations.
(2) In Section 3.3, the proportions of metals in the DOC had an identifying number, but it is unclear if the number is mass to mass or mole to mole. Metals having different molecular weights may be identified in different ways. For example, in reference 20 Kikuchi et al., (2017), reference 52 Baken et al., (2011), as well as many other authors, such as Amery et al., (2007, 2008), Chon et al., (2017) and Hsieh et al., (2019) μmol/g-OC has been used to represent the proportion of metal in the DOC. The same unit is easy to compare. In addition, in this article, these proportions were not compared to values in other surface water studies. Please express what the number unit is and show a comparison to other surface water studies.
(3) In Figures 6 and 7, the correlation of metal affinity with optical indicators is unclear and difficult to understand. The authors used an uncommon way to present the correlations between the two parameters. Please clarify the correlation.
(4) Does R2 have a negative value? Please explain.

Reviewer 3 Report
The work carried out in this research project is nicely represented. Hence I recommend this work for publication in this journal. However, the discussion section should be enriched by the addition of some latest references.
Reviewer 4 Report
The manuscript presents very interesting research results. Assessing the role of organic matter and related nutrients in the trace metal cycle is very important for understanding their behaviour in the aquatic environment. However, I have some comments about this manuscript.
First of all, I think that it should be reworded. The text structure should be improved because the reader is lost in information. The Authors should rethink how to present the results. Using tables and other figures, charts with clearly shown correlations and dependencies would make the article legible.
The discussion should be a separate chapter in which the authors discuss the results and how they can be interpreted in the perspective of previous research and working hypotheses. There should be no references to literature in the Results chapter (they are now).
In the manuscript there is no information about the mineral composition of the lacustrine sediments but there are discussions about it.
There is also no information on the lithology of rocks building the lake catchment area, which has an impact on the chemistry of groundwater and surface waters.
The Materials and Methods chapter does not explain all the procedures discussed in the results, e.g. see line 207.
Not all abbreviations are explained. They should be explained in the chapter Materials and Methods. E.g. M1, CDOM etc.
Figure 1 should contain a map with the location of the research area in the background of China
Round 2
Reviewer 4 Report
The manuscript has been improved. But I have one more remark. In the "Study Area" chapter, there is still no information on the lithology of rocks that build the catchment area. Information that they are metamorphic or clastic rocks (described in Fig. 1) is not sufficient. It should be describe exactly what these rocks are (limestones, gneisses ...?).